# On Riemannian Optimization over Positive Definite Matrices with the Bures-Wasserstein Geometry

**Andi Han**[1], **Bamdev Mishra**[2], **Pratik Jawanpuria**[2], **Junbin Gao**[1]
[1]The University of Sydney, Australia   [2]Microsoft, India
{andi.han, junbin.gao}@sydney.edu.au.
{bamdevm, pratik.jawanpuria}@microsoft.com.

## Abstract

In this paper, we comparatively analyze the Bures-Wasserstein (BW) geometry with the popular Affine-Invariant (AI) geometry for Riemannian optimization on the symmetric positive definite (SPD) matrix manifold. Our study begins with an observation that the BW metric has a linear dependence on SPD matrices in contrast to the quadratic dependence of the AI metric. We build on this to show that the BW metric is a more suitable and robust choice for several Riemannian optimization problems over ill-conditioned SPD matrices. We show that the BW geometry has a non-negative curvature, which further improves convergence rates of algorithms over the non-positively curved AI geometry. Finally, we verify that several popular cost functions, which are known to be geodesic convex under the AI geometry, are also geodesic convex under the BW geometry. Extensive experiments on various applications support our findings.

## 1 Introduction

Learning on symmetric positive definite (SPD) matrices is a fundamental problem in various machine learning applications, including metric and kernel learning [69, 24, 38, 13, 66], medical imaging [57, 56], natural language processing [37, 40], computer vision [28, 35, 34], multi-task learning [39, 54], domain adaptation [48, 41], modeling time-varying data [18], object detection [70], and quantum mechanics [53, 47], among others.

The set of SPD matrices of size $n \times n$, defined as $\mathbb{S}_{++}^n := \{\mathbf{X} : \mathbf{X} \in \mathbb{R}^{n \times n}, \mathbf{X}^\top = \mathbf{X}, \text{ and } \mathbf{X} \succ \mathbf{0}\}$, has a smooth manifold structure with a richer geometry than the Euclidean space. When endowed with a metric (inner product structure), the set of SPD matrices becomes a Riemannian manifold [11]. Hence, numerous existing works [57, 7, 42, 36, 34, 46, 56] have studied and employed the Riemannian optimization framework for learning over the space of SPD matrices [2, 16].

Several Riemannian metrics on $\mathbb{S}_{++}^n$ have been proposed such as the Affine-Invariant [57, 11], the Log-Euclidean [7, 58], the Log-Det [64, 20], and the Log-Cholesky [46], to name a few. One can additionally obtain different families of Riemannian metrics on $\mathbb{S}_{++}^n$ by appropriate parameterizations based on the principles of invariance and symmetry [23, 20, 67, 56]. However, to the best of our knowledge, a systematic study comparing the different metrics for optimizing generic cost functions defined on $\mathbb{S}_{++}^n$ is missing. Practically, the Affine-Invariant (AI) metric seems to be the most widely used metric in Riemannian first- and second-order algorithms (e.g., steepest descent, conjugate gradients, and trust regions) as it is the only Riemannian SPD metric available in manifold optimization toolboxes, such as Manopt [17], Manopt.jl [10], Pymanopt [68], ROPTLIB [32], and McTorch [50]. Moreover, many interesting problems in machine learning are found to be geodesic convex (generalization of Euclidean convexity) under the AI metric, which allows fast convergence of optimization algorithms [76, 31].

Recent works have studied the Bures-Wasserstein (BW) distance on SPD matrices [49, 12, 72]. It is a well-known result that the Wasserstein distance between two multivariate Gaussian densities is a function of the BW distance between their covariance matrices. Indeed, the BW metric is a Riemannian metric. Under this metric, the necessary tools for Riemannian optimization, including the Riemannian gradient and Hessian expressions, can be efficiently computed [49]. Hence, it is a promising candidate for Riemannian optimization on $\mathbb{S}^n_{++}$.

In this work, we theoretically and empirically analyze the quality of optimization with the BW geometry and show that it is a viable alternative to the default choice of AI geometry. Our analysis discusses the classes of cost functions (e.g., polynomial) for which the BW metric has better convergence rates than the AI metric. We also discuss cases (e.g., log-det) where the reverse is true. In particular, our contributions are as follows.

- We observe that the BW metric has a linear dependence on SPD matrices while the AI metric has a quadratic dependence. We show this impacts the condition number of the Riemannian Hessian and makes the BW metric more suited to learning ill-conditioned SPD matrices than the AI metric.

- In contrast to the non-positively curved AI geometry, the BW geometry is shown to be non-negatively curved, which leads to a tighter trigonometry distance bound and faster convergence rates for optimization algorithms.

- For both metrics, we analyze the convergence rates of Riemannian steepest descent and trust region methods and highlight the issues arising from the differences in the curvature and condition number of the Riemannian Hessian.

- We verify that common optimization problems that are geodesic convex under the AI metric are also geodesic convex under the BW metric.

- We support our analysis with extensive experiments on applications such as weighted least squares, trace regression, metric learning, and Gaussian mixture model.

## 2  Preliminaries

The fundamental ingredients for Riemannian optimization are Riemannian metric, exponential map, Riemannian gradient, and Riemannian Hessian. We refer readers to [2, 16] for a general treatment on Riemannian optimization.

A Riemannian metric is a smooth, bilinear, and symmetric positive definite function on the tangent space $T_x\mathcal{M}$ for any $x \in \mathcal{M}$. That is, $g : T_x\mathcal{M} \times T_x\mathcal{M} \to \mathbb{R}$, which is often written as an inner product $\langle \cdot, \cdot \rangle_x$. The induced norm of a tangent vector $u \in T_x\mathcal{M}$ is given by $\|u\|_x = \sqrt{\langle u, u \rangle_x}$. A geodesic on a manifold $\gamma : [0, 1] \to \mathcal{M}$ is defined as a locally shortest curve with zero acceleration. For any $x \in \mathcal{M}, u \in T_x\mathcal{M}$, the exponential map, $\mathrm{Exp}_x : T_x\mathcal{M} \to \mathcal{M}$ is defined such that there exists a geodesic curve $\gamma$ with $\gamma(0) = x$, $\gamma(1) = \mathrm{Exp}_x(u)$ and $\gamma'(0) = u$.

**First-order geometry and Riemannian steepest descent.**  Riemannian gradient of a differentiable function $f : \mathcal{M} \to \mathbb{R}$ at $x$, denoted as $\mathrm{grad}f(x)$, is a tangent vector that satisfies for any $u \in T_x\mathcal{M}$, $\langle \mathrm{grad}f(x), u \rangle_x = \mathrm{D}_u f(x)$, where $\mathrm{D}_u f(x)$ is the directional derivative of $f(x)$ along $u$. The Riemannian steepest descent method [71] generalizes the standard gradient descent in the Euclidean space to Riemannian manifolds by ensuring that the updates are along the geodesic and stay on the manifolds. That is, $x_{t+1} = \mathrm{Exp}_{x_t}(-\eta_t\,\mathrm{grad}f(x_t))$ for some step size $\eta_t$.

**Second-order geometry and Riemannian trust region.**  Second-order methods such as trust region and cubic regularized Newton methods are generalized to Riemannian manifolds [1, 3]. They make use of the Riemannian Hessian, $\mathrm{Hess}f(x) : T_x\mathcal{M} \to T_x\mathcal{M}$, which is a linear operator that is defined as the covariant derivative of the Riemannian gradient. Both the trust region and cubic regularized Newton methods are Hessian-free in the sense that only evaluation of the Hessian acting on a tangent vector, i.e., $\mathrm{Hess}f(x)[u]$ is required. Similar to the Euclidean counterpart, the Riemannian trust region method approximates the Newton step by solving a subproblem, i.e.,

$$\min_{u \in T_{x_t}\mathcal{M}:\|u\|_{x_t}\leq\Delta} m_{x_t}(u) = f(x_t) + \langle \mathrm{grad}f(x_t), u \rangle_{x_t} + \frac{1}{2}\langle \mathcal{H}_{x_t}[u], u \rangle_{x_t},$$

Table 1: Riemannian optimization ingredients for AI and BW geometries.

| | Affine-Invariant | Bures-Wasserstein |
|---|---|---|
| R.Metric | $g_{\mathrm{ai}}(\mathbf{U}, \mathbf{V}) = \mathrm{tr}(\mathbf{X}^{-1}\mathbf{U}\mathbf{X}^{-1}\mathbf{V})$ | $g_{\mathrm{bw}}(\mathbf{U}, \mathbf{V}) = \frac{1}{2}\mathrm{tr}(\mathcal{L}_{\mathbf{X}}[\mathbf{U}]\mathbf{V})$ |
| R.Exp | $\mathrm{Exp}_{\mathrm{ai},\mathbf{X}}(\mathbf{U}) = \mathbf{X}^{1/2}\exp(\mathbf{X}^{-1}\mathbf{U})\mathbf{X}^{1/2}$ | $\mathrm{Exp}_{\mathrm{bw},\mathbf{X}}(\mathbf{U}) = \mathbf{X} + \mathbf{U} + \mathcal{L}_{\mathbf{X}}[\mathbf{U}]\mathbf{X}\mathcal{L}_{\mathbf{X}}[\mathbf{U}]$ |
| R.Gradient | $\mathrm{grad}_{\mathrm{ai}}f(\mathbf{X}) = \mathbf{X}\nabla f(\mathbf{X})\mathbf{X}$ | $\mathrm{grad}_{\mathrm{bw}}f(\mathbf{X}) = 4\{\nabla f(\mathbf{X})\mathbf{X}\}_{\mathrm{S}}$ |
| R.Hessian | $\mathrm{Hess}_{\mathrm{ai}}f(\mathbf{X})[\mathbf{U}] = \mathbf{X}\nabla^2 f(\mathbf{X})[\mathbf{U}]\mathbf{X} + \{\mathbf{U}\nabla f(\mathbf{X})\mathbf{X}\}_{\mathrm{S}}$ | $\mathrm{Hess}_{\mathrm{bw}}f(\mathbf{X})[\mathbf{U}] = 4\{\nabla^2 f(\mathbf{X})[\mathbf{U}]\mathbf{X}\}_{\mathrm{S}} + 2\{\nabla f(\mathbf{X})\mathbf{U}\}_{\mathrm{S}} + 4\{\mathbf{X}\{\mathcal{L}_{\mathbf{X}}[\mathbf{U}]\nabla f(\mathbf{X})\}_{\mathrm{S}}\}_{\mathrm{S}} - \{\mathcal{L}_{\mathbf{X}}[\mathbf{U}]\mathrm{grad}_{\mathrm{bw}}f(\mathbf{X})\}_{\mathrm{S}}$ |

where $\mathcal{H}_{x_t} : T_{x_t}\mathcal{M} \to T_{x_t}\mathcal{M}$ is a symmetric and linear operator that approximates the Riemannian Hessian. $\Delta$ is the radius of trust region, which may be increased or decreased depending on how model value $m_{x_t}(u)$ changes. The subproblem is solved iteratively using a truncated conjugate gradient algorithm. The next iterate is given by $x_{t+1} = \mathrm{Exp}_{x_t}(u)$ with the optimized $u$.

Next, the eigenvalues and the condition number of the Riemannian Hessian are defined as follows, which we use for analysis in Section 3.

**Definition 1.** The minimum and maximum eigenvalues of $\mathrm{Hess}f(x)$ are defined as $\lambda_{\min} = \min_{\|u\|_x^2=1}\langle \mathrm{Hess}f(x)[u], u\rangle_x$ and $\lambda_{\max} = \max_{\|u\|_x^2=1}\langle \mathrm{Hess}f(x)[u], u\rangle_x$. The condition number of $\mathrm{Hess}f(x)$ is defined as $\kappa(\mathrm{Hess}f(x)) := \lambda_{\max}/\lambda_{\min}$.

**Function classes on Riemannian manifolds.** For analyzing algorithm convergence, we require the definitions for several important function classes on Riemannian manifolds, including geodesic convexity and smoothness. Similarly, we require the definition for geodesic convex sets that generalize (Euclidean) convex sets to manifolds [65, 74].

**Definition 2** (Geodesic convex set [65, 74]). A set $\mathcal{X} \subseteq \mathcal{M}$ is geodesic convex if for any $x, y \in \mathcal{X}$, the distance minimizing geodesic $\gamma$ joining the two points lies entirely in $\mathcal{X}$.

Indeed, this notion is well-defined for any manifold because a sufficiently small geodesic ball is always geodesic convex.

**Definition 3** (Geodesic convexity [65, 74]). Consider a geodesic convex set $\mathcal{X} \subseteq \mathcal{M}$. A function $f : \mathcal{X} \to \mathbb{R}$ is called geodesic convex if for any $x, y \in \mathcal{X}$, the distance minimizing geodesic $\gamma$ joining $x$ and $y$ satisfies $\forall t \in [0,1], f(\gamma(t)) \leq (1-t)f(x) + tf(y)$. Function $f$ is strictly geodesic convex if the equality holds only when $t = 0, 1$.

**Definition 4** (Geodesic strong convexity and smoothness [65, 33]). Under the same settings in Definition 3. A twice-continuously differentiable function $f : \mathcal{X} \to \mathbb{R}$ is called geodesic $\mu$-strongly convex if for any distance minimizing geodesic $\gamma$ in $\mathcal{X}$ with $\|\gamma'(0)\| = 1$, it satisfies $\frac{d^2 f(\gamma(t))}{dt^2} \geq \mu$, for some $\mu > 0$. Function $f$ is called geodesic $L$-smooth if $\frac{d^2 f(\gamma(t))}{dt^2} \leq L$, for some $L > 0$.

## 3  Comparing BW with AI for Riemannian optimization

This section starts with an observation of a linear-versus-quadratic dependency between the two metrics. From this observation, we analyze the condition number of the Riemannian Hessian. Then, we further compare the sectional curvature of the two geometries. Together with the differences in the condition number, this allows us to compare the convergence rates of optimization algorithms on the two geometries. We conclude this section by showing geodesic convexity of several generic cost functions under the BW geometry. The proofs for this section are in our extended report [27].

**AI and BW geometries on SPD matrices.** When endowed with a Riemannian metric $g$, the set of SPD matrices of size $n$ becomes a Riemannian manifold $\mathcal{M} = (\mathbb{S}_{++}^n, g)$. The tangent space at $\mathbf{X}$ is $T_{\mathbf{X}}\mathcal{M} := \{\mathbf{U} : \mathbf{U} \in \mathbb{R}^{n \times n} \text{ and } \mathbf{U}^\top = \mathbf{U}\}$. Under the AI and BW metrics, the Riemannian exponential map, Riemannian gradient, and Hessian are compared in Table 1, where we denote $\{\mathbf{A}\}_{\mathrm{S}} := (\mathbf{A} + \mathbf{A}^\top)/2$ and $\exp(\mathbf{A})$ as the matrix exponential of $\mathbf{A}$. $\mathcal{L}_{\mathbf{X}}[\mathbf{U}]$ is the solution to the matrix linear system $\mathcal{L}_{\mathbf{X}}[\mathbf{U}]\mathbf{X} + \mathbf{X}\mathcal{L}_{\mathbf{X}}[\mathbf{U}] = \mathbf{U}$ and is known as the Lyapunov operator. We use

$\nabla f(\mathbf{X})$ and $\nabla^2 f(\mathbf{X})$ to represent the first-order and second-order derivatives, i.e., the Euclidean gradient and Hessian, respectively. The derivations in Table 1 can be found in [56, 12]. In the rest of the paper, we use $\mathcal{M}_{\mathrm{ai}}$ and $\mathcal{M}_{\mathrm{bw}}$ to denote the SPD manifolds under the two metrics. From Table 1, the computational costs for evaluating the AI and BW ingredients are dominated by the matrix exponential/inversion operations and the Lyapunov operator $\mathcal{L}$ computation, respectively. Both at most cost $\mathcal{O}(n^3)$, which implies a comparable per-iteration cost of optimization algorithms between the two metric choices. This claim is validated in Section 4.

**A key observation.** From Table 1, the Affine-Invariant metric on the SPD manifold can be rewritten as for any $\mathbf{U}, \mathbf{V} \in T_\mathbf{X}\mathcal{M}$,

$$\langle \mathbf{U}, \mathbf{V} \rangle_{\mathrm{ai}} = \mathrm{tr}(\mathbf{X}^{-1}\mathbf{U}\mathbf{X}^{-1}\mathbf{V}) = \mathrm{vec}(\mathbf{U})^\top (\mathbf{X} \otimes \mathbf{X})^{-1}\mathrm{vec}(\mathbf{V}), \quad (1)$$

where $\mathrm{vec}(\mathbf{U})$ and $\mathrm{vec}(\mathbf{V})$ are the vectorizations of $\mathbf{U}$ and $\mathbf{V}$, respectively. Note that we omit the subscript $\mathbf{X}$ for inner product $\langle \cdot, \cdot \rangle$ to simplify the notation. The specific tangent space where the inner product is computed should be clear from contexts.

The Bures-Wasserstein metric is rewritten as, for any $\mathbf{U}, \mathbf{V} \in T_\mathbf{X}\mathcal{M}$,

$$\langle \mathbf{U}, \mathbf{V} \rangle_{\mathrm{bw}} = \frac{1}{2}\mathrm{tr}(\mathcal{L}_\mathbf{X}[\mathbf{U}]\mathbf{V}) = \frac{1}{2}\mathrm{vec}(\mathbf{U})^\top (\mathbf{X} \oplus \mathbf{X})^{-1}\mathrm{vec}(\mathbf{V}), \quad (2)$$

where $\mathbf{X} \oplus \mathbf{X} = \mathbf{X} \otimes \mathbf{I} + \mathbf{I} \otimes \mathbf{X}$ is the Kronecker sum.

**Remark 1.** Comparing Eq. (1) and (2) reveals that the BW metric has a linear dependence on $\mathbf{X}$ while the AI metric has a quadratic dependence. This suggests that optimization algorithms under the AI metric should be more sensitive to the condition number of $\mathbf{X}$ compared to the BW metric.

The above observation serves as a key motivation for further analysis.

### 3.1 Condition number of Riemannian Hessian at optimality

Throughout the rest of the paper, we make the following assumptions.

**Assumption 1.** (a). $f$ is at least twice continuously differentiable with a non-degenerate local minimizer $\mathbf{X}^*$. (b). The subset $\mathcal{X} \subseteq \mathcal{M}$ (usually as a neighbourhood of a center point) we consider throughout this paper is totally normal, i.e., the exponential map is a diffeomorphism.

Assumption 1 is easy to satisfy. Particularly, Assumption 1(b) is guaranteed for the SPD manifold under the AI metric because its geodesic is unique. Under the BW metric, for a center point $\mathbf{X}$, we can choose the neighbourhood such that $\mathcal{X} = \{\mathrm{Exp}_\mathbf{X}(\mathbf{U}) : \mathbf{I} + \mathcal{L}_\mathbf{X}[\mathbf{U}] \in \mathbb{S}^n_{++}\}$ as in [49]. In other words, $\mathcal{X}$ is assumed to be unique-geodesic under both the metrics.

We now formalize the impact of the linear-versus-quadratic dependency, highlighted in Remark 1. At a local minimizer $\mathbf{X}^*$ where the Riemannian gradient vanishes, we first simplify the expression for the Riemannian Hessian in Table 1.

On $\mathcal{M}_{\mathrm{ai}}$, $\langle \mathrm{Hess}_{\mathrm{ai}} f(\mathbf{X}^*)[\mathbf{U}], \mathbf{U} \rangle_{\mathrm{ai}} = \mathrm{tr}(\nabla^2 f(\mathbf{X}^*)[\mathbf{U}]\mathbf{U}) = \mathrm{vec}(\mathbf{U})^\top \mathbf{H}(\mathbf{X}^*)\mathrm{vec}(\mathbf{U})$, with $\mathbf{H}(\mathbf{X}) \in \mathbb{R}^{n^2 \times n^2}$ is the matrix representation of the Euclidean Hessian $\nabla^2 f(\mathbf{X})$ and $\mathbf{U} \in T_{\mathbf{X}^*}\mathcal{M}_{\mathrm{ai}}$. The maximum eigenvalue of $\mathrm{Hess}_{\mathrm{ai}} f(\mathbf{X}^*)$ is then given by $\lambda^*_{\max} = \max_{\|\mathbf{U}\|^2_{\mathrm{ai}}=1} \mathrm{vec}(\mathbf{U})^\top \mathbf{H}(\mathbf{X}^*)\mathrm{vec}(\mathbf{U})$, where $\|\mathbf{U}\|^2_{\mathrm{ai}} = \mathrm{vec}(\mathbf{U})^\top (\mathbf{X}^* \otimes \mathbf{X}^*)^{-1}\mathrm{vec}(\mathbf{U})$. This is a generalized eigenvalue problem with the solution to be the maximum eigenvalue of $(\mathbf{X}^* \otimes \mathbf{X}^*)\mathbf{H}(\mathbf{X}^*)$. Similarly, $\lambda^*_{\min}$ corresponds to the minimum eigenvalue of $(\mathbf{X}^* \otimes \mathbf{X}^*)\mathbf{H}(\mathbf{X}^*)$.

On $\mathcal{M}_{\mathrm{bw}}$, $\langle \mathrm{Hess}_{\mathrm{bw}} f(\mathbf{X}^*)[\mathbf{U}], \mathbf{U} \rangle_{\mathrm{bw}} = \mathrm{vec}(\mathbf{U})^\top \mathbf{H}(\mathbf{X}^*)\mathrm{vec}(\mathbf{U})$ and the norm is $\|\mathbf{U}\|^2_{\mathrm{bw}} = \mathrm{vec}(\mathbf{U})^\top (\mathbf{X}^* \oplus \mathbf{X}^*)^{-1}\mathrm{vec}(\mathbf{U})$. Hence, the minimum/maximum eigenvalue of $\mathrm{Hess}_{\mathrm{bw}} f(\mathbf{X}^*)$ equals the minimum/maximum eigenvalue of $(\mathbf{X}^* \oplus \mathbf{X}^*)\mathbf{H}(\mathbf{X}^*)$.

Let $\kappa^*_{\mathrm{ai}} := \kappa(\mathrm{Hess}_{\mathrm{ai}} f(\mathbf{X}^*)) = \kappa((\mathbf{X}^* \otimes \mathbf{X}^*)\mathbf{H}(\mathbf{X}^*))$ and $\kappa^*_{\mathrm{bw}} := \kappa(\mathrm{Hess}_{\mathrm{bw}} f(\mathbf{X}^*)) = \kappa((\mathbf{X}^* \oplus \mathbf{X}^*)\mathbf{H}(\mathbf{X}^*))$. The following lemma bounds these two condition numbers.

**Lemma 1.** *For a local minimizer $\mathbf{X}^*$ of $f(\mathbf{X})$, the condition number of $\mathrm{Hess} f(\mathbf{X}^*)$ satisfies*

$$\kappa(\mathbf{X}^*)^2/\kappa(\mathbf{H}(\mathbf{X}^*)) \leq \kappa^*_{\mathrm{ai}} \leq \kappa(\mathbf{X}^*)^2\kappa(\mathbf{H}(\mathbf{X}^*))$$
$$\kappa(\mathbf{X}^*)/\kappa(\mathbf{H}(\mathbf{X}^*)) \leq \kappa^*_{\mathrm{bw}} \leq \kappa(\mathbf{X}^*)\kappa(\mathbf{H}(\mathbf{X}^*)).$$

It is clear that $\kappa_{\text{bw}}^* \leq \kappa_{\text{ai}}^*$ when $\kappa(\mathbf{H}(\mathbf{X}^*)) \leq \sqrt{\kappa(\mathbf{X}^*)}$. This is true for linear, quadratic, higher-order polynomial functions and in general holds for several machine learning optimization problems on the SPD matrices (discussed in Section 4).

**Case 1** (Condition number for linear and quadratic optimization). For a linear function $f(\mathbf{X}) = \text{tr}(\mathbf{X}\mathbf{A})$, its Euclidean Hessian matrix is $\mathbf{H}(\mathbf{X}) = \mathbf{0}$. For a quadratic function $f(\mathbf{X}) = \text{tr}(\mathbf{X}\mathbf{A}\mathbf{X}\mathbf{B})$ with $\mathbf{A}, \mathbf{B} \in \mathbb{S}_{++}^n$, $\mathbf{H}(\mathbf{X}) = \mathbf{A} \otimes \mathbf{B} + \mathbf{B} \otimes \mathbf{A}$. Therefore, $\kappa(\mathbf{H}(\mathbf{X}^*))$ is a constant and for ill-conditioned $\mathbf{X}^*$, we have $\kappa(\mathbf{H}(\mathbf{X}^*)) \leq \sqrt{\kappa(\mathbf{X}^*)}$, which leads to $\kappa_{\text{ai}}^* \geq \kappa_{\text{bw}}^*$.

**Case 2** (Condition number for higher-order polynomial optimization). For an integer $\alpha \geq 3$, consider a function $f(\mathbf{X}) = \text{tr}(\mathbf{X}^\alpha)$ with derived $\mathbf{H}(\mathbf{X}) = \alpha \sum_{l=0}^{\alpha-2} (\mathbf{X}^l \otimes \mathbf{X}^{\alpha-l-2})$. We get $\kappa_{\text{ai}}^* = \alpha \sum_{l=1}^{\alpha-1} ((\mathbf{X}^*)^l \otimes (\mathbf{X}^*)^{\alpha-l})$ and $\kappa_{\text{bw}}^* = \alpha(\mathbf{X}^* \oplus \mathbf{X}^*)(\sum_{l=0}^{\alpha-2} (\mathbf{X}^l \otimes \mathbf{X}^{\alpha-l-2}))$. It is apparent that $\kappa_{\text{ai}}^* = \mathcal{O}(\kappa(\mathbf{X}^*)^\alpha)$ while $\kappa_{\text{bw}}^* = \mathcal{O}(\kappa(\mathbf{X}^*)^{\alpha-1})$. Hence, for ill-conditioned $\mathbf{X}^*$, $\kappa_{\text{ai}}^* \geq \kappa_{\text{bw}}^*$.

One counter-example where $\kappa_{\text{bw}}^* \geq \kappa_{\text{ai}}^*$ is the log-det function.

**Case 3** (Condition number for log-det optimization). For the log-det function $f(\mathbf{X}) = -\log\det(\mathbf{X})$, its Euclidean Hessian is $\nabla^2 f(\mathbf{X})[\mathbf{U}] = \mathbf{X}^{-1}\mathbf{U}\mathbf{X}^{-1}$ and $\mathbf{H}(\mathbf{X}) = \mathbf{X}^{-1} \otimes \mathbf{X}^{-1}$. At a local minimizer $\mathbf{X}^*$, $\text{Hess}_{\text{ai}} f(\mathbf{X}^*)[\mathbf{U}] = \mathbf{U}$ with $\kappa_{\text{ai}}^* = 1$. While on $\mathcal{M}_{\text{bw}}$, we have $\kappa_{\text{bw}}^* = \kappa((\mathbf{X}^* \oplus \mathbf{X}^*)((\mathbf{X}^*)^{-1} \otimes (\mathbf{X}^*)^{-1})) = \kappa((\mathbf{X}^*)^{-1} \oplus (\mathbf{X}^*)^{-1}) = \kappa(\mathbf{X}^*)$. Therefore, $\kappa_{\text{ai}}^* \leq \kappa_{\text{bw}}^*$.

### 3.2 Sectional curvature and trigonometry distance bound

To study the curvature of $\mathcal{M}_{\text{bw}}$, we first show in Lemma 2, the existence of a matching geodesic between the Wasserstein geometry of zero-centered non-degenerate Gaussian measures and the BW geometry of SPD matrices. Denote the manifold of such Gaussian measures under the $L^2$-Wasserstein distance as $(\mathcal{N}_0(\boldsymbol{\Sigma}), \mathcal{W}_2)$ with $\boldsymbol{\Sigma} \in \mathbb{S}_{++}^n$.

**Lemma 2.** *For any $\mathbf{X}, \mathbf{Y} \in \mathbb{S}_{++}^n$, a geodesic between $\mathcal{N}_0(\mathbf{X})$ and $\mathcal{N}_0(\mathbf{Y})$ on $(\mathcal{N}_0(\boldsymbol{\Sigma}), \mathcal{W}_2)$ is given by $\mathcal{N}_0(\gamma(t))$, where $\gamma(t)$ is the geodesic between $\mathbf{X}$ and $\mathbf{Y}$ on $\mathcal{M}_{\text{bw}}$.*

The following lemma builds on a result from the Wasserstein geometry [6] and uses Lemma 2 to analyze the sectional curvature of $\mathcal{M}_{\text{bw}}$.

**Lemma 3.** *$\mathcal{M}_{\text{bw}}$ is an Alexandrov space with non-negative sectional curvature.*

It is well-known that $\mathcal{M}_{\text{ai}}$ is a non-positively curved space [21, 56] while, in Lemma 3, we show that $\mathcal{M}_{\text{bw}}$ is non-negatively curved. The difference affects the curvature constant in the trigonometry distance bound of Alexandrov space [77]. This bound is crucial in analyzing convergence for optimization algorithms on Riemannian manifolds [77, 76]. In Section 3.3, only local convergence to a minimizer $\mathbf{X}^*$ is analyzed. Therefore, it suffices to consider a neighbourhood $\Omega$ around $\mathbf{X}^*$. In such a compact set, the sectional curvature is known to be bounded and we denote the lower bound as $K^-$.

The following lemma compares the trigonometry distance bounds under the AI and BW geometries. This bound was originally introduced for Alexandrov space with lower bounded sectional curvature [77]. The result for non-negatively curved spaces has been applied in many work [76, 60, 25] though without a formal proof. We show the proof in [27], where it follows from the Toponogov comparison theorem [52] on the unit hypersphere and Assumption 1.

**Lemma 4.** *Let $\mathbf{X}, \mathbf{Y}, \mathbf{Z} \in \Omega$, which forms a geodesic triangle on $\mathcal{M}$. Denote $x = d(\mathbf{Y}, \mathbf{Z}), y = d(\mathbf{X}, \mathbf{Z}), z = d(\mathbf{X}, \mathbf{Y})$ as the geodesic side lengths and let $\theta$ be the angle between sides $y$ and $z$ such that $\cos(\theta) = \langle \text{Exp}_{\mathbf{X}}^{-1}(\mathbf{Y}), \text{Exp}_{\mathbf{X}}^{-1}(\mathbf{Z}) \rangle / (yz)$. Then, we have*

$$x^2 \leq \zeta y^2 + z^2 - 2yz\cos(\theta),$$

*where $\zeta$ is a curvature constant. Under the AI metric, $\zeta = \zeta_{\text{ai}} = \frac{\sqrt{|K_{\text{ai}}^-|}D}{\tanh(\sqrt{|K_{\text{ai}}^-|}D)}$ with $D$ as the diameter bound of $\Omega$, i.e. $\max_{\mathbf{X}_1, \mathbf{X}_2 \in \Omega} d(\mathbf{X}, \mathbf{Y}) \leq D$. Under the BW metric, $\zeta = \zeta_{\text{bw}} = 1$.*

It is clear that $\zeta_{\text{ai}} > \zeta_{\text{bw}} = 1$, which leads to a tighter bound under the BW metric.

### 3.3 Convergence analysis

We now analyze the local convergence properties of the Riemannian steepest descent and trust region methods under the two Riemannian geometries. Convergence is established in terms of the

Riemannian distance induced from the geodesics. We begin by presenting a lemma that shows in a neighbourhood of $\mathbf{X}^*$, the second-order derivatives of $f \circ \mathrm{Exp}_{\mathbf{X}}$ are both lower and upper bounded.

**Lemma 5.** *In a totally normal neighbourhood $\Omega$ around a non-degenerate local minimizer $\mathbf{X}^*$, for any $\mathbf{X} \in \Omega$, it satisfies that $\lambda_{\min}^*/\alpha \leq \frac{d^2}{dt^2} f(\mathrm{Exp}_{\mathbf{X}}(t\mathbf{U})) \leq \alpha\lambda_{\max}^*$, for some $\alpha \geq 1$ and $\|\mathbf{U}\| = 1$. $\lambda_{\max}^* > \lambda_{\min}^* > 0$ are the largest and smallest eigenvalues of $\mathrm{Hess} f(\mathbf{X}^*)$.*

For simplicity of the analysis, we assume such an $\alpha$ is universal under both the Riemannian geometries. We, therefore, can work with a neighbourhood $\Omega$ with diameter uniformly bounded by $D$, where we can choose $D := \min\{D_{\mathrm{ai}}, D_{\mathrm{bw}}\}$ such that $\alpha$ is universal.

One can readily check that under Lemma 5 the function $f$ is both $\mu$-geodesic strongly convex and $L$-geodesic smooth in $\Omega$ where $\mu = \lambda_{\min}^*/\alpha$ and $L = \alpha\lambda_{\max}^*$. We now present the local convergence analysis of the two algorithms, which are based on results in [77, 1].

**Theorem 1** (Local convergence of Riemannian steepest descent). *Under Assumption 1 and consider a non-degenerate local minimizer $\mathbf{X}^*$. For a neighbourhood $\Omega \ni \mathbf{X}^*$ with diameter bounded by $D$ on two Riemannian geometries $\mathcal{M}_{\mathrm{ai}}, \mathcal{M}_{\mathrm{bw}}$, running Riemannian steepest descent from $\mathbf{X}_0 \in \Omega$ with a fixed step size $\eta = \frac{1}{\alpha\lambda_{\max}^*}$ yields for $t \geq 2$,*

$$d^2(\mathbf{X}_t, \mathbf{X}^*) \leq \alpha^2 D^2 \kappa^* \left(1 - \min\{\frac{1}{\zeta}, \frac{1}{\alpha^2\kappa^*}\}\right)^{t-2} .$$

**Theorem 2** (Local convergence of Riemannian trust region). *Under the same settings as in Theorem 1, assume further in $\Omega$, it holds that (1) $\|\mathcal{H}_{\mathbf{X}_t} - \mathrm{Hess} f(\mathbf{X}_t)\| \leq \ell\|\mathrm{grad} f(\mathbf{X}_t)\|$ and (2) $\|\nabla^2(f \circ \mathrm{Exp}_{\mathbf{X}_t})(\mathbf{U}) - \nabla^2(f \circ \mathrm{Exp}_{\mathbf{X}_t})(\mathbf{0})\| \leq \rho\|\mathbf{U}\|$ for some $\ell, \rho$ universal on $\mathcal{M}_{\mathrm{ai}}, \mathcal{M}_{\mathrm{bw}}$. Then running Riemannian trust region from $\mathbf{X}_0 \in \Omega$ yields,*

$$d(\mathbf{X}_t, \mathbf{X}^*) \leq (2\sqrt{\rho} + \ell)(\kappa^*)^2 d^2(\mathbf{X}_{t-1}, \mathbf{X}^*).$$

Theorems 1 and 2 show that $\mathcal{M}_{\mathrm{bw}}$ has a clear advantage compared to $\mathcal{M}_{\mathrm{ai}}$ for learning ill-conditioned SPD matrices where $\kappa_{\mathrm{bw}}^* \leq \kappa_{\mathrm{ai}}^*$. For first-order algorithms, $\mathcal{M}_{\mathrm{bw}}$ has an additional benefit due to its non-negative sectional curvature. As $\zeta_{\mathrm{ai}} > \zeta_{\mathrm{bw}} = 1$, the convergence rate degrades on $\mathcal{M}_{\mathrm{ai}}$. Although the convergence is presented in Riemannian distance, it can be readily converted to function value gap by noticing $\frac{\mu}{2} d^2(\mathbf{X}_t, \mathbf{X}^*) \leq f(\mathbf{X}_t) - f(\mathbf{X}^*) \leq \frac{L}{2} d^2(\mathbf{X}_t, \mathbf{X}^*)$. Additionally, we note that these local convergence results hold regardless of whether the function is geodesic convex or not, and similar comparisons also exist for other Riemannian optimization methods.

### 3.4 Geodesic convexity under BW metric for cost functions of interest

Finally we show geodesic convexity of common optimization problems on $\mathcal{M}_{\mathrm{bw}}$. Particularly, we verify that linear, quadratic, log-det optimization, and also certain geometric optimization problems, that are geodesic convex under the AI metric, are also geodesic convex under the BW metric.

**Proposition 1.** *For any $\mathbf{A} \in \mathbb{S}_+^n$, where $\mathbb{S}_+^n := \{\mathbf{Z} : \mathbf{Z} \in \mathbb{R}^{n \times n}, \mathbf{Z}^\top = \mathbf{Z}, \text{ and } \mathbf{Z} \succeq \mathbf{0}\}$, the functions $f_1(\mathbf{X}) = \mathrm{tr}(\mathbf{X}\mathbf{A})$, $f_2(\mathbf{X}) = \mathrm{tr}(\mathbf{X}\mathbf{A}\mathbf{X})$, and $f_3(\mathbf{X}) = -\log\det(\mathbf{X})$ are geodesic convex on $\mathcal{M}_{\mathrm{bw}}$.*

Based on the result in Proposition 1, we also prove geodesic convexity of a reparameterized version of the Gaussian density estimation and mixture model on $\mathcal{M}_{\mathrm{bw}}$ (discussed in Section 4). Similar claims on $\mathcal{M}_{\mathrm{ai}}$ can be found in [31].

We further show that monotonic functions on sorted eigenvalues are geodesic convex on $\mathcal{M}_{\mathrm{bw}}$. This is an analogue of [65, Theorem 2.3] on $\mathcal{M}_{\mathrm{ai}}$.

**Proposition 2.** *Let $\lambda^\downarrow : \mathbb{S}_{++}^n \to \mathbb{R}_+^n$ be the decreasingly sorted eigenvalue map and $h : \mathbb{R}_+ \to \mathbb{R}$ be an increasing and convex function. Then $f(\mathbf{X}) = \sum_{j=1}^k h(\lambda_j^\downarrow(\mathbf{X}))$ for $1 \leq k \leq n$ is geodesic convex on $\mathcal{M}_{\mathrm{bw}}$. Examples of such functions include $f_1(\mathbf{X}) = \mathrm{tr}(\exp(\mathbf{X}))$ and $f_2(\mathbf{X}) = \mathrm{tr}(\mathbf{X}^\alpha)$, $\alpha \geq 1$.*

## 4 Experiments

In this section, we compare the empirical performance of optimization algorithms under different Riemannian geometries for various problems. In addition to AI and BW, we also include the Log-Euclidean (LE) geometry [7] in our experiments.

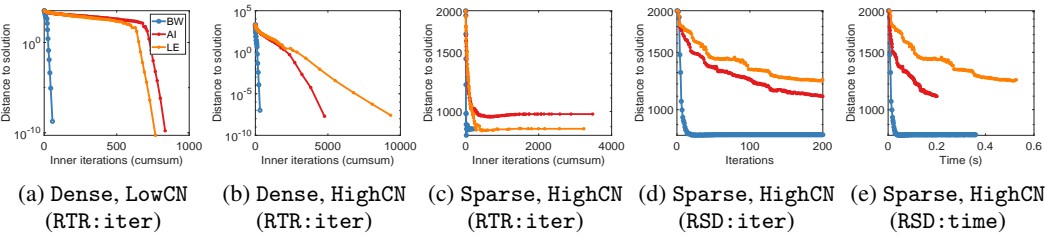

|(a) Dense, LowCN
(RTR:iter)|(b) Dense, HighCN
(RTR:iter)|(c) Sparse, HighCN
(RTR:iter)|(d) Sparse, HighCN
(RSD:iter)|(e) Sparse, HighCN
(RSD:time)|

Figure 1: Weighted least squares problem.

The LE geometry explores the the linear space of symmetric matrices where the matrix exponential acts as a global diffeomorphism from the space to $\mathbb{S}_{++}^n$. The LE metric is defined as

$$\langle \mathbf{U}, \mathbf{V} \rangle_{\mathrm{le}} = \mathrm{tr}(\mathrm{D}_{\mathbf{U}} \log(\mathbf{X}) \mathrm{D}_{\mathbf{V}} \log(\mathbf{X})) \tag{3}$$

for any $\mathbf{U}, \mathbf{V} \in T_{\mathbf{X}} \mathcal{M}$, where $\mathrm{D}_{\mathbf{U}} \log(\mathbf{X})$ is the directional derivative of matrix logarithm at $\mathbf{X}$ along $\mathbf{U}$. Following [69, 51, 58], for deriving various Riemannian optimization ingredients under the LE metric (3), we consider the parameterization $\mathbf{X} = \exp(\mathbf{S})$, where $\mathbf{S} \in \mathbb{S}^n$, i.e., the space of $n \times n$ symmetric matrices. Equivalently, optimization on the SPD manifold with the LE metric is identified with optimization on $\mathbb{S}^n$ and the function of interest becomes $f(\exp(\mathbf{S}))$ for $\mathbf{S} \in \mathbb{S}^n$. While the Riemannian gradient can be computed efficiently by exploiting the directional derivative of the matrix exponential [4], deriving the Riemannian Hessian is tricky and we rely on finite-difference Hessian approximations [15].

We present convergence mainly in terms of the distance to the solution $\mathbf{X}^*$ whenever applicable. The distance is measured in the Frobenius norm, i.e., $\|\mathbf{X}_t - \mathbf{X}^*\|_{\mathrm{F}}$. When $\mathbf{X}^*$ is not known, convergence is shown in the modified Euclidean gradient norm $\|\mathbf{X}_t \nabla f(\mathbf{X}_t)\|_{\mathrm{F}}$. This is comparable across different metrics as the optimality condition $\mathbf{X}^* \nabla f(\mathbf{X}^*) = \mathbf{0}$ arises from problem structure itself [43]. We initialize the algorithms with the identity matrix for the AI and BW metrics and zero matrix for the LE metric (i.e., the matrix logarithm of the identity).

We mainly present the results on the Riemannian trust region (RTR) method, which is the method of choice for Riemannian optimization. Note for RTR, the results are shown against the cumulative sum of inner iterations (which are required to solve the trust region subproblem at every iteration). We also include the Riemannian steepest descent (RSD) and Riemannian stochastic gradient (RSGD) [14] methods for some examples. The experiments are conducted in Matlab using the Manopt toolbox [17] on a i5-10500 3.1GHz CPU processor.

In our extended report [27], we include additional experiments comparing convergence in objective function values for the three geometries. We also present results for the Riemannian conjugate gradient method, and results with different initializations (other than the identity and zero matrices) to further support our claims.

The code can be found at `https://github.com/andyjm3/AI-vs-BW`.

**Weighted least squares.** We first consider the weighted least squares problem with the symmetric positive definite constraint. The optimization problem is $\min_{\mathbf{X} \in \mathbb{S}_{++}^n} f(\mathbf{X}) = \frac{1}{2} \|\mathbf{A} \odot \mathbf{X} - \mathbf{B}\|_{\mathrm{F}}^2$, which is encountered in for example, SPD matrix completion [63] where $\mathbf{A}$ is a sparse matrix. The Euclidean gradient and Hessian are $\nabla f(\mathbf{X}) = (\mathbf{A} \odot \mathbf{X} - \mathbf{B}) \odot \mathbf{A}$ and $\nabla^2 f(\mathbf{X})[\mathbf{U}] = \mathbf{A} \odot \mathbf{U} \odot \mathbf{A}$, respectively. Hence, at optimal $\mathbf{X}^*$, the Euclidean Hessian in matrix representation is $\mathbf{H}(\mathbf{X}^*) = \mathrm{diag}(\mathrm{vec}(\mathbf{A} \odot \mathbf{A}))$. We experiment with two choices of $\mathbf{A}$, i.e. $\mathbf{A} = \mathbf{1}_n \mathbf{1}_n^\top$ (Dense) and $\mathbf{A}$ as a random sparse matrix (Sparse). The former choice for $\mathbf{A}$ leads to well-conditioned $\mathbf{H}(\mathbf{X}^*)$ while the latter choice leads to an ill-conditioned $\mathbf{H}(\mathbf{X}^*)$. Also note that when $\mathbf{A} = \mathbf{1}_n \mathbf{1}_n^\top$, $\kappa_{\mathrm{ai}}^* = \kappa(\mathbf{X}^*)^2$ and $\kappa_{\mathrm{bw}}^* = \kappa(\mathbf{X}^*)$.

We generate $\mathbf{X}^*$ as a SPD matrix with size $n = 50$ and exponentially decaying eigenvalues. We consider two cases with condition numbers $\kappa(\mathbf{X}^*) = 10$ (LowCN) and $10^3$ (HighCN). The matrix $\mathbf{B}$ is generated as $\mathbf{B} = \mathbf{A} \odot \mathbf{X}^*$. Figure 1 compares both RSD and RTR for different metrics. When $\mathbf{A}$ is either dense or sparse, convergence is significantly faster on $\mathcal{M}_{\mathrm{bw}}$ than both $\mathcal{M}_{\mathrm{ai}}$ and $\mathcal{M}_{\mathrm{le}}$. The advantage of using $\mathcal{M}_{\mathrm{bw}}$ becomes more prominent in the setting when condition number of $\mathbf{X}^*$ is high. Figure 1(e) shows that $\mathcal{M}_{\mathrm{bw}}$ is also superior in terms of runtime.

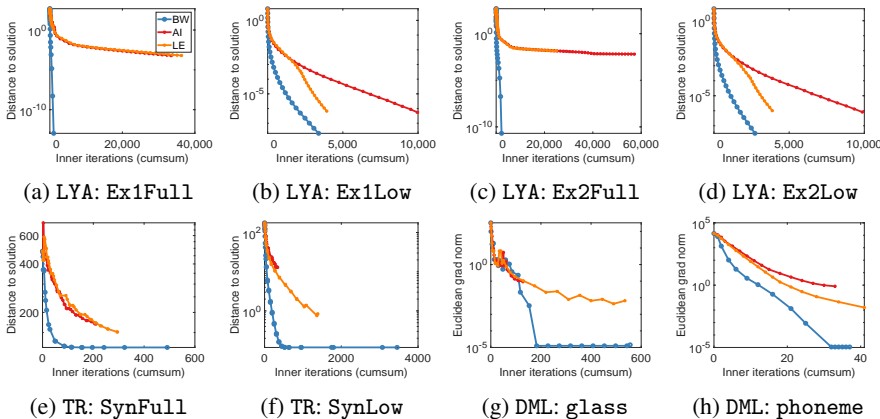

(a) LYA: Ex1Full  (b) LYA: Ex1Low  (c) LYA: Ex2Full  (d) LYA: Ex2Low

(e) TR: SynFull  (f) TR: SynLow  (g) DML: glass  (h) DML: phoneme

Figure 2: Lyapunov equation (a, b, c, d), trace regression (e, f), and metric learning (g, h) problems.

**Lyapunov equations.** Continuous Lyapunov matrix equation, $\mathbf{AX} + \mathbf{XA} = \mathbf{C}$ with $\mathbf{X} \in \mathbb{S}_{++}^n$, is commonly employed in analyzing optimal control systems and differential equations [59, 45]. When $\mathbf{A}$ is stable, i.e., $\lambda_i(\mathbf{A}) > 0$ and $\mathbf{C} \in \mathbb{S}_{++}^n$, the solution $\mathbf{X}^* \succ \mathbf{0}$ and is unique [44]. When $\mathbf{C} \in \mathbb{S}_+^n$ and is low rank, $\mathbf{X}^* \in \mathbb{S}_+^n$ is also low rank. We optimize the following problem for solving the Lyapunov equation [73], i.e., $\min_{\mathbf{X} \in \mathbb{S}_{++}^n} f(\mathbf{X}) = \mathrm{tr}(\mathbf{XAX}) - \mathrm{tr}(\mathbf{XC})$. The Euclidean gradient and Hessian are respectively $\nabla f(\mathbf{X}) = \mathbf{AX} + \mathbf{XA} - \mathbf{C}$ and $\nabla^2 f(\mathbf{X})[\mathbf{U}] = \mathbf{AU} + \mathbf{UA}$ with $\mathbf{H}(\mathbf{X}) = \mathbf{A} \oplus \mathbf{A}$. At optimal $\mathbf{X}^*$, the condition number $\kappa(\mathbf{H}(\mathbf{X}^*)) = \kappa(\mathbf{A})$.

We experiment with two settings for the matrix $\mathbf{A}$, i.e. $\mathbf{A}$ as the Laplace operator on the unit square where we generate 7 interior points so that $n = 49$ (Ex1), and $\mathbf{A}$ is a particular Toeplitz matrix with $n = 50$ (Ex2). The generated $\mathbf{A}$ matrices are ill-conditioned. The above settings correspond to Examples 7.1 and 7.3 in [45]. Under each setting, $\mathbf{X}^*$ is set to be either full or low rank. The matrix $\mathbf{C}$ is generated as $\mathbf{C} = \mathbf{AX}^* + \mathbf{X}^*\mathbf{A}$. The full rank $\mathbf{X}^*$ is generated from the full-rank Wishart distribution while the low rank $\mathbf{X}^*$ is a diagonal matrix with $r = 10$ ones and $n - r$ zeros in the diagonal. We label the four cases as Ex1Full, Ex1Low, Ex2Full, and Ex2Low. The results are shown in Figures 2(a)-(d), where we observe that in all four cases, the BW geometry outperforms both AI and LE geometries.

**Trace regression.** We consider the regularization-free trace regression model [62] for estimating covariance and kernel matrices [61, 19]. The optimization problem is $\min_{\mathbf{X} \in \mathbb{S}_{++}^d} f(\mathbf{X}) = \frac{1}{2m} \sum_{i=1}^m (\mathbf{y}_i - \mathrm{tr}(\mathbf{A}_i^\top \mathbf{X}))^2$, where $\mathbf{A}_i = \mathbf{a}_i \mathbf{a}_i^\top$, $i = 1, ..., m$ are some rank-one measurement matrices. Thus, we have $\nabla f(\mathbf{X}) = \sum_{i=1}^m (\mathbf{a}_i^\top \mathbf{X} \mathbf{a}_i - \mathbf{y}_i) \mathbf{A}_i$ and $\nabla^2 f(\mathbf{X})[\mathbf{U}] = \sum_{i=1}^m (\mathbf{a}_i^\top \mathbf{U} \mathbf{a}_i) \mathbf{A}_i$.

We create $\mathbf{X}^*$ as a rank-$r$ Wishart matrix and $\{\mathbf{A}_i\}$ as rank-one Wishart matrices and generate $\mathbf{y}_i = \mathrm{tr}(\mathbf{A}_i \mathbf{X}^*) + \sigma \epsilon_i$ with $\epsilon_i \sim \mathcal{N}(0, 1)$, $\sigma = 0.1$. We consider two choices, $(m, d, r) = (1000, 50, 50)$ and $(1000, 50, 10)$, which are respectively labelled as SynFull and SynLow. From Figures 2(e)&(f), we also observe that convergence to the optimal solution is faster for the BW geometry.

**Metric learning.** Distance metric learning (DML) aims to learn a distance function from samples and a popular family of such distances is the Mahalanobis distance, i.e. $d_{\mathbf{M}}(\mathbf{x}, \mathbf{y}) = \sqrt{(\mathbf{x} - \mathbf{y})^\top \mathbf{M}(\mathbf{x} - \mathbf{y})}$ for any $\mathbf{x}, \mathbf{y} \in \mathbb{R}^d$. The distance is parameterized by a symmetric positive semi-definite matrix $\mathbf{M}$. We refer readers to this survey [66] for more discussions on this topic. We particularly consider a logistic discriminant learning formulation [24]. Given a training sample $\{\mathbf{x}_i, y_i\}_{i=1}^N$, denote the link $t_{ij} = 1$ if $y_i = y_j$ and $t_{ij} = 0$ otherwise. The objective is given by $\min_{\mathbf{M} \in \mathbb{S}_{++}^d} f(\mathbf{M}) = -\sum_{i,j} (t_{ij} \log p_{ij} + (1 - t_{ij}) \log(1 - p_{ij}))$, with $p_{ij} = (1 + \exp(d_{\mathbf{M}}(\mathbf{x}_i, \mathbf{x}_j)))^{-1}$. We can derive the matrix Hessian as $\mathbf{H}(\mathbf{M}) = \sum_{i,j} p_{ij}(1 - p_{ij})(\mathbf{x}_i - \mathbf{x}_j)(\mathbf{x}_i - \mathbf{x}_j)^\top \otimes (\mathbf{x}_i - \mathbf{x}_j)(\mathbf{x}_i - \mathbf{x}_j)^\top$. Notice $\kappa(\mathbf{H}(\mathbf{M}^*))$ depends on $\mathbf{M}^*$ only through the constants $p_{ij}$. Thus, the condition number will not be much affected by $\kappa(\mathbf{M}^*)$.

We consider two real datasets, glass and phoneme, from the Keel database [5]. The number of classes is denoted as $c$. The statistics of these two datasets are $(N, d, c) = (241, 9, 7)$ for glass

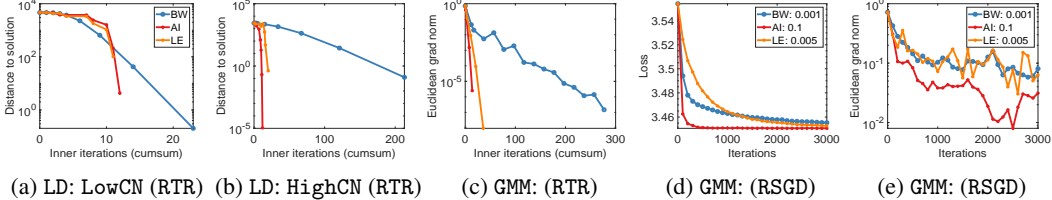

(a) LD: LowCN (RTR)  (b) LD: HighCN (RTR)  (c) GMM: (RTR)  (d) GMM: (RSGD)  (e) GMM: (RSGD)

Figure 3: Log-det maximization (a, b) and Gaussian mixture model (c, d, e) problems.

$(5404, 5, 2)$ for `phoneme`. In Figures 2(g)&(h), we similarly see the advantage of using the BW metric compared to the other two metrics that behave similarly.

**Log-det maximization.** As discussed in Section 3.1, log-det optimization is one instance where $\kappa_{\mathrm{bw}}^* \geq \kappa_{\mathrm{ai}}^*$. We first consider minimizing negative log-determinant along with a linear function as studied in [75]. That is, for some $\mathbf{C} \in \mathbb{S}_{++}^n$, the objective is $\min_{\mathbf{X} \in \mathbb{S}_{++}^n} f(\mathbf{X}) = \mathrm{tr}(\mathbf{X}\mathbf{C}) - \log \det(\mathbf{X})$. The Euclidean gradient and Hessian are given by $\nabla f(\mathbf{X}) = \mathbf{C} - \mathbf{X}^{-1}$ and $\nabla^2 f(\mathbf{X})[\mathbf{U}] = \mathbf{X}^{-1}\mathbf{U}\mathbf{X}^{-1}$. This problem is geodesic convex under both AI and BW metrics. We generate $\mathbf{X}^*$ the same way as in the example of weighted least square with $n = 50$ and set $\mathbf{C} = (\mathbf{X}^*)^{-1}$. We consider two cases with condition number cn = 10 (`LowCN`) and $10^3$ (`HighCN`). As expeted, we observe faster convergence of AI and LE metrics over the BW metric in Figures 3(a)&(b). This is even more evident when the condition number increases.

**Gaussian mixture model.** Another notable example of log-det optimization is the Gaussian density estimation and mixture model problem. Following [31], we consider a reformulated problem on augmented samples $\mathbf{y}_i^\top = [\mathbf{x}_i^\top; 1], i = 1, ..., N$ where $\mathbf{x}_i \in \mathbb{R}^d$ are the original samples. The density is parameterized by the augmented covariance matrix $\boldsymbol{\Sigma} \in \mathbb{R}^{d+1}$. Notice that the log-likelihood of Gaussian is geodesic convex on $\mathcal{M}_{\mathrm{ai}}$, but not on $\mathcal{M}_{\mathrm{bw}}$. We, therefore, define $\mathbf{S} = \boldsymbol{\Sigma}^{-1}$ and the reparameterized log-likelihood is $p_{\mathcal{N}}(\mathbf{Y}; \mathbf{S}) = \sum_{i=1}^N \log \left((2\pi)^{1-d/2} \exp(1/2) \det(\mathbf{S})^{1/2} \exp(-\frac{1}{2}\mathbf{y}_i^\top \mathbf{S}\mathbf{y}_i)\right)$, which is now geodesic convex on $\mathcal{M}_{\mathrm{bw}}$ due to Proposition 1. Hence, we can solve the problem of Gaussian mixture model similar as in [31].

Here, we test on a dataset included in the MixEst package [30]. The dataset has 1580 samples in $\mathbb{R}^2$ with 3 Gaussian components. In Figure 3(c), we observe a similar pattern with RTR as in the log-det example. We also include performance of RSGD, which is often preferred for large scale problems. We set the batch size to be 50 and consider a decaying step size, with the best initialized step size shown in Figures 3(d)&(e). Following [8], the algorithms are initialized with `kmeans++`. We find that the AI geometry still maintains its advantage under the stochastic setting.

## 5 Conclusion and discussion

In this paper, we show that the less explored Bures-Wasserstein (BW) geometry for SPD matrices should often be the preferable choice than the Affine-Invariant geometry for optimization, particularly for learning ill-conditioned matrices. A systematic analysis shows that the BW metric preserves geodesic convexity of some popular cost functions and leads to better convergence rates in optimization.

We also theoretically discuss a 'counter-example' of log-det optimization where the Affine-Invariant (AI) geometry enjoys a better second-order conditioning and validate our findings empirically. This issue is addressed in our recent work [26], where we propose a generalized Bures-Wasserstein (GBW) geometry that is built on a generalization of the Lyapunov operator in the metric:

$$\langle \mathbf{U}, \mathbf{V} \rangle_{\mathrm{gbw}} = \frac{1}{2}\mathrm{vec}(\mathbf{U})^\top (\mathbf{X} \otimes \mathbf{M} + \mathbf{M} \otimes \mathbf{X})^{-1}\mathrm{vec}(\mathbf{V}), \tag{4}$$

where $\mathbf{U}$ and $\mathbf{V}$ are symmetric matrices and $\mathbf{M}$ is a given SPD matrix. When $\mathbf{M} = \mathbf{I}$, the metric (4) reduces to the special BW metric (2). The use of the parameter $\mathbf{M}$ in (4) allows great flexibility for optimization algorithms. For one, choosing a particular $\mathbf{M}$ allows preconditioning the Hessian by locally approximating the AI geometry. This leads to improved convergence for log-det optimization.

Our comparisons between AI and BW geometries are based on optimization over generic cost functions. For specific problems, however, there may exist other alternative metrics that potentially work better. This is an interesting research direction to pursue. We also remark that optimization is not the only area where the AI and BW geometries can be compared. It would be useful to qualitatively compare the two metrics for other learning problems on SPD matrices, such as barycenter learning. In addition, kernel methods have been studied on the SPD manifold [42, 29, 78] that embed SPD matrices to a high dimensional feature space, known as the Reproducing Kernel Hilbert Space (RKHS). Such representations are used for subsequent learning tasks such as clustering or classification. But only a positive definite kernel provides a valid RKHS. We show in appendix Section A that the induced Gaussian kernel based on the BW distance is a positive definite kernel unlike the case for the AI metric. This difference highlights a potential advantage of the BW metric for representation learning on SPD matrices.

## A Positive definite kernel from BW geometry

In this section, we show the existence of a positive definite Gaussian kernel on $\mathcal{M}_{\mathrm{bw}}$. Existing works [55, 22] have studied the Wasserstein distance kernel. First, we present the definition of a positive (resp. negative) definite function as in [9].

**Definition 5.** Consider $\mathcal{X}$ to be a nonempty set. A function $f : \mathcal{X} \times \mathcal{X} \to \mathbb{R}$ is called positive definite if and only if $k$ is symmetric and for all integers $m \geq 2$, $\{x_1, ..., x_m\} \subseteq \mathcal{X}$ and $\{c_1, ..., c_m\} \subseteq \mathbb{R}$, it satisfies $\sum_{i,j=1}^{m} c_i c_j f(x_i, x_j) \geq 0$. A function $f$ is called negative definite if and only if under the same conditions, it satisfies $\sum_{i,j=1}^{m} c_i c_j f(x_i, x_j) \leq 0$ with $\sum_{i=1}^{m} c_i = 0$.

The following theorem shows the Gaussian kernel induced from the BW distance is positive definite on the SPD manifold.

**Theorem 3.** *The induced Gaussian kernel* $k(\cdot, \cdot) := \exp(-d_{\mathrm{bw}}^2(\cdot, \cdot)/(2\sigma^2))$ *is positive definite.*

*Proof of Theorem 3.* From Theorem 4.3 in [42], it suffices to prove the BW distance $d_{\mathrm{bw}}^2$ is negative definite. Indeed for any $m \geq 2$, $\{\mathbf{X}_1, ..., \mathbf{X}_m\} \subseteq \mathcal{M}_{\mathrm{bw}}$, $\{c_1, ..., c_m\} \subseteq \mathbb{R}$ with $\sum_{i=1}^{m} c_i = 0$, we have

$$
\sum_{i,j=1}^{m} c_i c_j d_{\mathrm{bw}}^2(\mathbf{X}_i, \mathbf{X}_j)
$$
$$
= \sum_{j=1}^{m} c_j \sum_{i=1}^{m} c_i \mathrm{tr}(\mathbf{X}_i) + \sum_{i=1}^{m} c_i \sum_{j=1}^{m} c_j \mathrm{tr}(\mathbf{X}_j) - 2 \sum_{i,j=1}^{m} c_i c_j \mathrm{tr}(\mathbf{X}_i^{1/2} \mathbf{X}_j \mathbf{X}_i^{1/2})^{1/2}
$$
$$
= -2 \sum_{i,j=1}^{m} c_i c_j \mathrm{tr}(\mathbf{X}_i^{1/2} \mathbf{X}_j \mathbf{X}_i^{1/2})^{1/2} \leq 0.
$$

This shows $d_{\mathrm{bw}}^2$ is negative definite, and therefore, the exponentiated Gaussian kernel is positive definite. □

## Broader Impact

This paper is of a theoretical nature and does not foresee any immediate societal impacts.

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
