# OpenReview forum: "On Riemannian Optimization over Positive Definite Matrices with the Bures-Wasserstein Geometry"
_NeurIPS.cc/2021/Conference — NeurIPS 2021 Poster_

### Official Review · Reviewer_crTV · 2021-07-12

**Rating:** 8
**Confidence:** 4

**Summary:**

This paper investigates the advantages of the Bures-Wasserstein (BW) metric over the widely used Affine Invariant (AI) one, for the manifold of symmetric positive definite (SPD) matrices. Rewriting the two metrics in a compatible manner reveals that for a wide variety of objectives, the BW metric is a linear function of the input matrix whereas the AI has a quadratic dependence. One exception is the log-det function which is included in the analysis. The paper experimentally validates that for ill-conditioned problems BW is a better choice compared to AI.

**Limitations And Societal Impact:**

This paper has a theoretical nature and does not have direct negative impacts on society. The limitations are explained in the discussions section.

**Main Review:**

PROS

- SPD matrices are used ubiquitously in computer vision. Hence, the fundamental contribution of this paper is expected to have a significant impact in the field -- at least for the portion which consider optimization on SPDs.
- I like the principled approach of the paper. It starts with an analysis of the problem and later fills in some theoretical gaps in the literature.
- The experiments include various relevant problems and reflect what the theory stands for: BW does indeed converge better.
- The content, albeit involved, is clearly presented with adequate references. I would be happy to see this paper at NeurIPS.

CONS

While I have not gone through all the derivations, I have not spotted major weaknesses of the paper. Here are some minor points of concern:
- The fact that BW provides a valid RKHS is very important. In fact the non-positive definiteness of the kernel has already caused nuisances for computer vision people. See for example:
* Birdal, T., Arbel, M., Simsekli, U., & Guibas, L. J. (2020). Synchronizing probability measures on rotations via optimal transport. In Proceedings of the IEEE/CVF Conference on Computer Vision and Pattern Recognition (pp. 1569-1579). *
Therefore, I would be happy to see further discussions on the connections with kernel methods here.
- Can we extend this study and compare with other metrics? Are there other metrics which also show linear dependence but somewhat inferior to BW?
- Log-det seems to be just one counter-example. I feel like there are many other functions of this type. Could we have a general characterization of when affinte-invariant metric is a good choice and when not? Maybe a general recipe towards choosing a metric?
- While the analysis is conducted for an analytical Hessian, in practice the Hessian is approximated using a finite differencing scheme. Does that create a significant gap with the theory? Can we actually study/quantify this gap? Why would the paper not use auto-differentiation to be exact?
- The paper assumes the availability of an exponential map. For SPDs this is okay. But for conducting a similar analysis on other manifolds, a (maybe first-order) retraction should be considered. Is it possible to extend the analysis to the case of retractions?
- I suggest that the findings here are incorporated into Manopt.

**Time Spent Reviewing:**

4

---

> ### Author Response · Authors · 2021-08-09
> **Response to Reviewer crTV**
>
> Thank you for the feedback. Below are our responses to your questions.
>
> Q1. $\textit{Including discussion on BW providing valid RKHS}$
>
> A1. Thanks for this suggestion. Yes, the result is indeed interesting, and hence, we included a discussion on that in supplementary. We will extend this discussion on kernel methods in the final version.
>
> Q2. $\textit{Extending analysis to other metrics.}$
>
> A2. Similar analysis on other metrics can be performed but not without difficulties. It is more trickier without analytical Hessian expressions. For example, study on geodesic convexity would be involved as well. Furthermore, considering the computational aspects, other Riemannian metrics may not be competitive than AI, LE, and BW.
>
> Q3. $\textit{More functions where AI is a good choice.}$
>
> A3. Indeed, there can be more functions besides log-det where AI is preferred in terms of condition number. But given its quadratic dependence of condition number on $X^*$ compared to the linear dependence of BW, there may not be that many functions with better conditioning under AI metric.
>
> Q4. $\textit{Analysis with approximated Hessian.}$
>
> A4. Regarding Hessian approximation, as long as the difference between the approximated Hessian and the true Hessian is bounded, there should not be a significant gap with the theory. This can be a good extension to the theory and we intend to study it.

---

### Official Review · Reviewer_w1Wx · 2021-07-14

**Rating:** 6
**Confidence:** 4

**Summary:**

This paper provides several results about the Bures-Wasserstein  (BW) geometry on the positive definite matrices. A thorough comparison with the Affine Invariant (AI) metric is made. The main focus of the paper is on optimization: is it faster to optimize a function under the AI or under the BW metric? The authors argue that when the solution of the problem $X^*$ is ill-conditioned, then algorithms derived under BW metric have faster convergence theoretical convergence. This is due to the conditioning of the Riemannian Hessian at the optimum, which is better for the BW metric when $X^*$ has poor conditioning. The authors also demonstrate that a variety of functions that are convex for the AI geometry are also convex for the BW geometry.
Experiments on an array of synthetic problems validate that both Riemannian gradient descent and Riemannian trust region methods are faster in the BW geometry.

**Limitations And Societal Impact:**

The authors seem over optimistic about the performance of BW metric compared to AI metric.

**Main Review:**

**Originality** :  This paper derives some novel results about the Bures-Wasserstein geometry, and provides some insights comparing it to the affine-invariant metric. Such comparisons are important because both metrics are widely used in practice. To the best of my knowledge, insights regarding the conditioning (lemma 1), distance bounds (lemma 4) or convexity (prop 1) are novel.
- Lemma 2 is already known, see Example 1.7 in [1]
- I think that lemma 4 and Lemma 5 in [2] are similar.

**Quality** : I think that comparing in a thorough manner the convergence properties of optimization algorithms under different metrics is an important and interesting problem. However, I have some concerns regarding the results of the authors. In particular, I think that the authors paint a picture that is too biased in favor the BW metric, and that there are some flaws in the analysis.

- One of the most important result of the paper is that the conditioning of the Hessian under the BW metric, $\kappa_{BW}$, is smaller than the conditioning under the AI metric, $\kappa_{AI}$, when $\kappa(H) \leq \sqrt{\kappa(X^*)}$, where $H$ is the Euclidean Hessian of the function, and $X^*$ is the solution. This means that algorithms using the BW geometry will be faster when $X^*$ is poorly conditioned, not when $X^*$ is well conditioned. Some parts of the text seem to skip this important condition (e.g. L163). Also, when $\kappa(H) \leq \sqrt{\kappa(X^*)}$, then lemma 1 gives $\kappa_{BW} \geq \kappa(X^*)/ \kappa(H) \geq \kappa(H)$. Therefore, BW has itself a worse conditioning that using simply the Euclidean metric ! This fact should be mentioned, and this also raises the question of using plain Euclidean algorithms (for instance, projected gradient descent) in the experiments.

- The constant $\alpha$ in Lemma 5 plays a key role in the convegence speed of the algorithms. However, I think it is important to insist that $\alpha$ depends on the metric used, since it is defined with the Exp. As a consequence, it is impossible to conclude that the rates in Thm.1 and 2 are better or worse for the BW/AI metric, even when $\kappa_{BW} < \kappa_{AI}$. In order to draw a conclusion, a study of the behavior of $\alpha$ under the two metrics should be carried.

- I also have concerns regarding the experiments. I replicated the least squares experiment with success. However, I tried another  related cost function, where here there is a matrix product instead of Hadamard product $f(X) = \frac12 \|AX - B\|$, where $B = AX^*$ with $X^*$ ill-conditioned, and I found that the two algorithms performed quite comparably, even when $X^*$ is poorly conditioned. I suspect that this is due to the constant $\alpha$ in the bounds. Also, I think that an experiment based on symmetric matrices taken from a real dataset would strengthen the point of the paper.




**Clarity**: the paper is well written and pleasant to read. It is also well organized.

**Significance** : as I already stated, I think that a clear and accurate comparison of the two metrics is a significant result that is of broad interest to any researcher working with PSD matrices.

**Misc** :
- Another field where PSD matrices with the AI/BW matrices are extensively used is EEG/MEG processing, with works such as [3]
- It should be mentioned that the paper assumes that the Euclidean gradient of $f$ is always symmetric.
- The actual BW and AI distances between two PSD matrices is never explicitly written in the paper, while these are the most used quantities. Including them in Table 1 would be worthwhile.
- Following the definition of conditioning of the authors, the conditioning of a linear problem is ill-defined.
- In lemma 1, are the inequalities tight?
- In Lemma 5, what is $t$?
- In fig.1 with gradient descent, how was chosen the step size? It is of crucial importance.
- In fig1. c, d, e, using a log scale for the y axis would make things more readable.

[1]: McCann, Robert J. "A convexity principle for interacting gases." Advances in mathematics 128.1 (1997): 153-179.

[2]: Hongyi Zhang and Suvrit Sra. First-order methods for geodesically convex optimization. In
Conference on Learning Theory, pages 1617–1638. PMLR, 2016.

[3]: Barachant, Alexandre, et al. "Multiclass brain–computer interface classification by Riemannian geometry." IEEE Transactions on Biomedical Engineering 59.4 (2011): 920-928.

**Time Spent Reviewing:**

10

---

> ### Author Response · Authors · 2021-08-09
> **Response to Reviewer w1Wx**
>
> Thank you for the feedback. Below are our responses.
>
> Q1. $\textit{Discuss well conditioned case analysis for BW and AI.}$
>
> A1. Lemma 1 provides bounds on condition numbers for BW and AI metric. This bound only claims when optimal solution is ill-conditioned, $\kappa_{bw} \leq \kappa_{ai}$. However, when $X^*$ is well-conditioned, we cannot conclude which condition number is smaller. This is problem-dependent. Having said that, the linear vs quadratic dependence of $\kappa_{bw}$ and $\kappa_{ai}$ comes into play in most cases. Our experiments empirically verify good performance of BW across both ill and well conditioned cases. We will include a discussion on this in the final version.
>
> Q2. $\textit{Case of Euclidean metric.}$
>
> A2. There are inherent difficulties associated with taking the Euclidean metric for Riemannian optimization (and it has been well studied in literature). For example, the geodesics are flat lines which do not guarantee positive definiteness of iterates. Limiting points of sequences are not guaranteed to be positive definite either. Also, projected gradient descent does not lead to positive definiteness (only semi-definiteness is ensured). Refer to many citations in the introduction. See also (Hosseini and Sra, 2020) and (Gao et al, 2020). Apart from condition number analysis where we compare metrics, we also consider geodesic convexity (Section 3.4), which allows many non-convex functions, e.g. GMM (Hosseini and Sra, 2020) to be formulated as geodesic convex.
>
>
>
> Q3. $\textit{Theorems 1, 2 depend on $\alpha$, which depends on metric used}$.
>
> A3. First, we emphasize that Theorem 2 does not depend on $\alpha$. Second, even for Theorem 1, for analysis, we can always choose an $\alpha$ that is universal in a sufficiently small neighbourhood $\Omega$ with a uniform diameter $D$ (see Lines 214-216). For *local* convergence analysis, the rate primarily depends on the condition number rather than $\alpha$.
>
>
>
> Q4. $\textit{``AI and BW perform comparably on } f = | AX - B |^2 \textit{for ill-conditioned } X^*.$''
>
> A4. No, this is not the case. We have tested on the matrix product cost $f(X) = \frac{1}{2}|| AX - B ||^2$  with $X^*$ ill-conditioned and find BW converging much faster than AI ! We suspect your implementation of the gradient and Hessian may not be correct. Please use checkgradient and checkhessian to see if your implementation is correct. The correct Euclidean gradient is $\nabla f(X) = \lbrace A^\top(AX-B) \rbrace_S$ and Hessian is $\nabla f(X)[U]=\lbrace A^\top A U \rbrace_S$ where we need to use the symmetric operator.
>
>
> Q5. $\textit{Real-data experiments based on ``symmetric'' matrices?}$
>
> A5. This comment is not clear to us. All our experiments deal with symmetric and positive definite matrices. Also, we have shown results for metric learning on 6 real datasets, "glass" ,"phoneme" in the main text and "Iris", "Balance", "Newthyroid", and "Popfailure" in supplementary.
>
>
> Q6. $\textit{``The paper assumes that the Euclidean gradient of}$ $f$ $\textit{is symmetric''}.$
>
> A6. No, we do not assume that in the paper. However, the Euclidean gradient is going to be symmetric anyway as the optimization variable ($X$) itself is symmetric (and positive definite).
>
> Q7. $\textit{``Conditioning definition for linear problem is ill-defined''.}$
>
> A7. No, that is not correct. The Riemannian Hessian of a linear problem exists (even when the Euclidean Hessian is zero) and is well defined. See, Table 1. Therefore, Definition 1 is valid for a linear problem.
>
>
>
> Q8. $\textit{Tightness of Lemma 1. Step size choice of gradient descent.}$
>
> A8. As far as we know, the inequalities are tight for the most general cases. The step size for gradient descent is selected via the backtracking line search. See Manopt.org for more descriptions.

---

> > ### Comment · Reviewer_w1Wx · 2021-08-17
> > **response**
> >
> > Dear authors,
> >
> > Thank you for addressing my concerns.
> >
> > A1: thanks.
> >
> > A2: Thanks for the clarification. I still believe that this important fact deserves some attention in future versions of the paper. Also, I am not particularly convinced by the argument about projection leading to semi-positive-definiteness. Indeed, since $S^n_{++}$ is an open set, and $S^n_+$ is the minimal closed set that contains it, the problem $\inf_{x\in S^n_{++}} f(x)$ is equivalent to $\inf_{x\in S^n_{+}} f(x)$ provided that $f$ is continous. In other words, if the minimizer is on the edge (has a $0$ eigenvalue), then $\min_{x\in S^n_{++}} f(x)$ has no solution, while $\min_{x\in S^n_{+}} f(x)$ has one (for instance, we can simply take $f(x) = |x|^2$). Therefore, an euclidean algorithm that optimizes on $S^n_+$ and not on $S^n_{++}$ should not be so problematic in practice: either the solution has a 0 eigenvalue, and then riemannian algorithms will fail anyway (they will try to go towards the border), or it is definite and then the euclidean algorithm solving on $S^n_+$ will recover it.
> >
> > A3. Sorry for misreading thm.2. Thanks for the clarification.
> >
> > A4. I apologize here: as the authors mention, **there was indeed a bug in my code**. Upon fixing it, I can observe the same behavior as in the paper: bw is indeed the faster method on this problem. Thanks for the clarification.
> >
> > A5 : I am sorry for the bad formulation; I was indeed asking for more real data experiments, with matrices coming from the real world.  I think that it might be worth it to put some of the examples in the supplementary to the main text, as it demonstrates the practical importance of the BW metric.
> >
> > A6: I think that, for the sale of clarity, it is important to precisely state that the ambient space for $f$ is the set of symmetric matrices, and that given the Euclidean gradient of $f$ is obtained by projecting the Euclidean gradient when the ambient space is $\mathbb{R}^{n\times n}$. The gradient at l283 should then be symmetrized (unless A and B are symmetric, which should be specified then).
> >
> > A7: I do not see how the Hessian does not cancel for the linear problem: L160-161, when $H=0$, the authors take $\kappa(0)$?
> >
> > A8: thanks
> >
> >
> >
> > Best,

---

> > > ### Author Response · Authors · 2021-08-18
> > > **Response to Reviewer w1Wx**
> > >
> > > Thank you for your responses. Specifically, on the linear problem and the Hessian at optimality. We should point out that Section 3.1 assumes $X^*$ to be a local minimizer as stated in Assumption 1 (Line 141-142). For the linear problem, there is no local minimizer as the objective function is unbounded in the open set. We will make this clear as well as incorporate your other suggestions in the final version.

---

> > > > ### Comment · Reviewer_w1Wx · 2021-08-18
> > > > **response**
> > > >
> > > > Thank you for addressing my concerns. I have raised my rating from 4 to 6.
> > > >
> > > > Best,

---

### Official Review · Reviewer_VX14 · 2021-07-15

**Rating:** 6
**Confidence:** 4

**Summary:**

This paper studies the Bures-Wasserstein (BW) geometry of symmetric positive-definite (SPD) matrices and compares it to the commonly-used affine-invariant (AI) geometry in the context of Riemannian optimization. It is shown theoretically that the BW geometry has several advantages. Particularly, it is shown that (i) the BW metric is linearly dependent on the SPD matrix, (ii) the condition number of the Riemannian Hessian of the BW metric is smaller than that of the AI metric, and (iii) the curvature constant of the BW metric is smaller than that of the AI metric. Based on these properties, it is shown that the convergence of Riemannian steepest descent and Riemannian trust-region based on the BW metric is faster than based on the AI metric. The theoretical results are demonstrated on simulations of six problems: weighted least squares, Lyapunov equations, trace regression, metric learning, log-det maximization, and Gaussian mixture model.

**Ethical Concerns:**

I do not have any ethical concerns.

**Limitations And Societal Impact:**

The limitations of the work could be better presented and discussed, especially since the paper considers only optimization properties. I do not foresee any potential negative societal impact.

**Main Review:**

The Riemannian geometry of SPD matrices has become popular recently, and the choice of geometry/metric is of paramount importance. This paper provides a systematic way to choose the metric from an optimization standpoint. As I outline below, I believe the paper has few drawbacks. However, despite its weaknesses, I believe this paper presents a solid contribution to the increasing literature and practice of Riemannian geometry of SPD matrices.

Detailed comments:
- The argument that a smaller condition number and/or smaller constant curvature is necessarily better is problematic. In terms of optimization, it is indeed shown for example that a smaller condition number translates to better convergence. However, if this is the only criterion, then one could claim that the Euclidean metric outperforms both the AI and the BW metrics. In other words, the paper claims that “simpler” metrics are better for optimization, but it does not consider the effect of the metric on the induced geometry.
- To make my comment above more concrete, I would love to see, in addition to the simulations, results on some real data, where the contribution of using Riemannian geometry is established, and where the choice of the metric also determines the “quality” of the analysis.
- In the experiments (line 265), the Frobenius (Euclidean) norm is used as a measure of distance to the desired solution. This departs from the entire message of the paper advocating for Riemannian geometry and could be particularly problematic when measuring the convergence rate.
- In many figures (e.g., Fig 1 (a)+(b)), it appears as if not all the graphs have reached convergence. Why do the graphs have different lengths?
- line 185: the geodesic gamma(t) is undefined (a missing reference to Prop 3 in the appendix).
- line 200: it would be better to explicitly define “geodesic triangle”, although it is implied from the context.

**After rebuttal:**
I thank the authors for their response to my comments.
After reading the other reviews and the authors' responses, my two concerns regarding the Euclidean metric and norm still remain. That said, I am increasing my score and recommend accepting the paper, since I believe it presents analysis and results that will be interesting to the community of people working on the Riemannian geometry of SPD matrices.

**Time Spent Reviewing:**

4

---

> ### Author Response · Authors · 2021-08-09
> **Response to Reviewer VX14**
>
> Thank you for your feedback and here are our responses to your concerns.
>
> Q1. $\textit{Condition number analysis only criterion for comparison? Claim on ``simple" metrics being better.}$
>
> A1. This is not a complete reading of the paper as we do not claim 'simpler' metrics are better for optimization. Apart from the condition number and curvature analysis, we also emphasize that induced geodesic convexity (section 3.4) is also an important criterion for Riemannian optimization. Indeed both BW and AI have rich geometries (section 3.2) that stand out and make them ideal candidates for optimization.
>
> Also, assessment of Euclidean metric being better based just on section 3.1 is not correct. There exist inherent difficulties and issues associated with Euclidean metric for Riemannian optimization, which have been well studied in literature. For example, the geodesics are flat lines which do not guarantee positive definiteness. Limiting points of sequences are not guaranteed to be positive definite either. Also, Euclidean methods like projected gradient descent do not lead to positive definiteness of iterates (only semi-definiteness is ensured).
>
>
>
> Q2. $\textit{Experiments on real data and quality analysis experiments.}$
>
> A2. We have shown results for metric learning  on 6 real datasets. "glass", "phoneme" in the main text and "Iris", "Balance", "Newthyroid", and "Popfailure" in supplementary.
>
> We also consider cases where convergence quality of different algorithms are shown. Figures 12, 13, 14, 15, 16 show algorithms converge to different optimal solutions, thereby emphasizing the role of metrics. See the figures with loss functions (shown in linear scale; in log scale convergence to different optimas are visible clearly, we would include them) and gradient norms.
>
>
>
>
>
> Q3. $\textit{Use of Frobenius norm as comapison criterion.}$
>
> A3. Notice that under different metrics, Riemannian distances are different. Hence, as common experiments benchmark, we use Frobenius norm. We mention this in Lines 266 - 267. Also, we also compare the convergence in terms of function value gap (included as supplementary).
>
>
>
> Q4. $\textit{Graphs have different lengths.}$
>
> A4. We set a fixed maximum number of iterations across metrics. For brevity, we only show figures up to a point where we can understand the behaviours clearly.

---

### Official Review · Reviewer_LtgQ · 2021-07-17

**Rating:** 6
**Confidence:** 3

**Summary:**

The authors consider the Riemannian optimization with Bures-Wasserstein (BW) geometry for symmetric positive definite (SPD) matrix manifold. The authors compare the proposed approach with its counterpart with the popular Affine-Invariant (AI) geometry.

The authors illustrate that the proposed approach (with BW geometry) is more suitable than its counterpart with AI geometry when SPD matrices are ill-conditioned.


**Limitations And Societal Impact:**

Yes

**Main Review:**

Riemannian optimization with SPD matrices is important for several approaches in machine learning. The authors consider the new raising Bures-Wasserstein geometry for this Riemannian optimization.

Overall, the paper is easy to follow and well-organized. The authors also evaluate the proposed approaches on several problems in machine learning to illustrate the theoretical findings.

By analyzing the condition number (Lemma 1), the authors show that BW geometry is more suitable for Riemannian optimization with SPD matrices than its counterpart with the popular AI geometry when SPD matrices are ill-conditioned.

I have some concerns as follow:
+ In Remark 1, from Equ. (1), why BM metric has a linear dependence on X? (but not its inverse?)
+ For Lemma 5, is there any assumption (e.g., the strongly convex and smooth) for the $f$ function?
+ It seems that results about convergence in Theorem 1 and 2 are straightforward results from the literature? It is better in case the authors give more discussions about what new results are here, comparing with the literature?
+ For Figure 2, row 1, (b) and (d), why the AI and LE has the same behaviors for some first iteration, but LE converges better with more iterations?
+ Although the authors perform several experiments, most of them are with synthesized data. It is better in case the authors consider more illustrations with real-world problem/data (such as with the metric learning).

~~~~~~~~
After the rebuttal:

I thank the authors for the response. I agree with the other reviewers that it is better to give general properties for the loss function when AI geometry is better (currently, only 1 example is listed in the submission). Overall, I keep leaning on the positive side.

**Time Spent Reviewing:**

8 hours

---

> ### Author Response · Authors · 2021-08-09
> **Response to Reviewer LtgQ**
>
> Thank you for the feedback. Below are our responses to your concerns.
>
> Q1. $\textit{Notion of linear dependence of BW metric.}$
>
> A1.  By linear dependence, we mean linear dependence of $K = X \oplus X$ on $X$, rather than $K^{-1}$. This is also mentioned in Section 3.1.
>
>
> Q2. $\textit{Strong convexity assumptions in Lemma 5?}$
>
> A2. No, there are no strong assumptions (eg strongly convex or smoothness) on
> $f$. This lemma follows from continuity of composed function $f \circ {\rm Exp}$. See proof details in supplementary.
>
>
> Q3. $\textit{Theorems 1 and 2 and their relations with existing results}$.
>
> A3. As mentioned in Line 219, Theorems 1 and 2 are modified from existing results. We restate the known results in terms of Riemannian distance and emphasize the dependence on condition number.
>
>
> Q4. $\textit{Figure 2 (b),(d), AI and LE behave similarly initially but LE converges better with more iterations}$.
>
> A4. Most optimization methods consume initial few iterations to find a good valley of attraction. Eg, TR method takes the few iterations to find the right TR radius. However, the dynamics of the algorithms are very different (as iterations progress).
>
>
> Q5. $\textit{Real-world datasets for metric learning.}$
>
> A5. We have shown results on 6 real datasets. "glass", "phoneme" in the main text and "Iris", "Balance", "Newthyroid", and "Popfailure" in supplementary.

---

### Decision · Program_Chairs · 2021-09-27

**Decision:**

Accept (Poster)

**Comment:**

The authors consider the Riemannian optimization with Bures-Wasserstein geometry for symmetric positive definite matrix manifold. Since the reviewers agree that the content of the paper will interest many researchers working on this subject, we conclude that the paper is worthy of acceptance. I think that the comment on the Frobenius norm by a reviewer is important. I expect that the authors add some comments on that issue in the revision.